# Divergent Patterns and Trends in Breast Cancer Incidence, Mortality and Survival Among Older Women in Germany and the United States

**DOI:** 10.3390/cancers12092419

**Published:** 2020-08-26

**Authors:** Lina Jansen, Bernd Holleczek, Klaus Kraywinkel, Janick Weberpals, Chloé Charlotte Schröder, Andrea Eberle, Katharina Emrich, Hiltraud Kajüter, Alexander Katalinic, Joachim Kieschke, Alice Nennecke, Eunice Sirri, Jörg Heil, Andreas Schneeweiss, Hermann Brenner

**Affiliations:** 1Division of Clinical Epidemiology and Aging Research, German Cancer Research Center (DKFZ), 69120 Heidelberg, Germany; janick.weberpals@gmail.com (J.W.); ccschroeder@gmx.de (C.C.S.); h.brenner@dkfz.de (H.B.); 2Saarland Cancer Registry, 66119 Saarbrücken, Germany; B.Holleczek@gbe-ekr.saarland.de; 3German Centre for Cancer Registry Data (ZfKD), Robert Koch-Institute, 13353 Berlin, Germany; KraywinkelK@rki.de; 4Cancer Registry of Bremen, Leibniz Institute for Prevention Research and Epidemiology—BIPS, 28359 Bremen, Germany; eberle@leibniz-bips.de; 5Cancer Registry of Rhineland-Palatinate, Institute for Medical Biostatistics, Epidemiology and Informatics, University Medical Center, Johannes Gutenberg University Mainz, 55116 Mainz, Germany; emrich@krebsregister-rlp.de; 6Cancer Registry of North Rhine-Westphalia, 44801 Bochum, Germany; Hiltraud.Kajueter@krebsregister.nrw.de; 7Cancer Registry of Schleswig-Holstein, 23552 Lübeck, Germany; alexander.katalinic@uksh.de; 8Cancer Registry of Lower Saxony, 26121 Oldenburg, Germany; kieschke@offis-care.de (J.K.); eunice.sirri@offis-care.de (E.S.); 9Hamburg Cancer Registry, 20539 Hamburg, Germany; Alice.Nennecke@bsg.hamburg.de; 10Department of Gynecology and Obstetrics, University Women’s Clinic, 69120 Heidelberg, Germany; joerg.heil@med.uni-heidelberg.de; 11National Center for Tumor Diseases, Division Gynecologic Oncology, University Hospital and German Cancer Research Center, 69120 Heidelberg, Germany; andreas.schneeweiss@nct-heidelberg.de; 12Division of Preventive Oncology, German Cancer Research Center (DKFZ), and National Center for Tumor Diseases (NCT), 69120 Heidelberg, Germany; 13German Cancer Consortium (DKTK), German Cancer Research Center, 69120 Heidelberg, Germany

**Keywords:** breast cancer, incidence, survival, mortality, Germany, United States, trends

## Abstract

Background: Breast cancer treatment has changed tremendously over the last decades. In addition, the use of mammography screening for early detection has increased strongly. To evaluate the impact of these developments, long-term trends in incidence, mortality, stage distribution and survival were investigated for Germany and the United States (US). Methods: Using population-based cancer registry data, long-term incidence and mortality trends (1975–2015), shifts in stage distributions (1998–2015), and trends in five-year relative survival (1979–2015) were estimated. Additionally, trends in five-year relative survival after standardization for stage were explored (2004–2015). Results: Age-standardized breast cancer incidence rates were much higher in the US than in Germany in all periods, whereas age-standardized mortality began to lower in the US from the 1990s on. The largest and increasing differences were observed for patients aged 70+ years with a 19% lower incidence but 45% higher mortality in Germany in 2015. For this age group, large differences in stage distributions were observed, with 29% (Germany) compared to 15% (US) stage III and IV patients. Age-standardized five-year relative survival increased strongly between 1979–1983 and 2013–2015 in Germany (+17% units) and the US (+19% units) but was 9% units lower in German patients aged 70+ years in 2013–2015. This difference was entirely explained by differences in stage distributions. Conclusions: Overall, our results are in line with a later uptake and less extensive utilization of mammography screening in Germany. Further studies and efforts are highly needed to further explore and overcome the increased breast cancer mortality among elderly women in Germany.

## 1. Introduction

Breast cancer (BC) is the most common cancer in women worldwide with an estimated 2.08 million new cancer cases diagnosed in 2018 [1]. It is estimated that 276,480 women will be newly diagnosed with BC and 42,170 women will die from BC in the United States in 2020 [2]. For Germany for 2016, 68,948 new BC diagnoses and 18,570 BC deaths were estimated [3]. In general, incidence rates vary strongly across countries and have increased over time [4]. The observed variations are mainly attributed to changes in reproductive patterns, the use of postmenopausal hormone therapy, and implementation of mammography screening programs [5,6,7]. Mortality rates have declined strongly over time in many countries. To what extent this mortality reduction can be attributed to mammography screening has been highly debated [8].

BC prognosis has improved steadily over time in most countries reaching a 5-year age-standardized survival in 2010–2014 between 70.8% (Russia) and 89.1% (Iceland) in Europe and 90.2% in the US [9]. In the US and in Germany, 5-year relative survival was 90% in 2015 [2] and 87% in 2015/16 [3], respectively. BC is a heterogeneous disease. Prognosis and treatment depend strongly on the cancer stage at diagnosis and the molecular subtype with the worst prognosis in stage IV and triple-negative BC [10]. Further classification methods based on genetic profiles are being developed to allow an even more accurate prognosis for recurrence and survival to further personalize treatment approaches [11,12]. In addition to classification and therapeutic improvements, changes in BC prognosis over time as well as variations across countries are affected by the implementation of mammography screening programs. Thus, improvements in BC survival at the population level do not directly reflect an improvement in BC care.

To get a thorough overview, incidence, mortality, and survival should be investigated in relation to each other [13]. Furthermore, a comparison of two or more countries might help in interpreting trends [13]. Here, we provide a comparison of long-term trends in BC incidence, mortality, and survival in Germany and the US, which were chosen for comparison because treatment guidelines are overall similar in both countries [14] but implementation and use of mammography screening differ strongly. An organized mammography screening program was implemented in Germany between 2005 and 2009 [15], while such screening programs were introduced in the US in the late 1980s [8]. Furthermore, while screening is recommended for patients aged 50–69 years in Germany, the US Preventive Services Task Force recommended screening without an upper age limit until 2009 when recommendations were limited to women below 75 years of age [16]. Thus, the comparisons of these two countries can provide important insights. 

## 2. Materials and Methods

### 2.1. Data Sources 

In Germany, cancer registration has been implemented on a federal state level and, therefore, the start and quality of cancer registration varies across these states. For years of diagnosis before around 1998, when only a few registries were implemented in Germany, data from the Saarland Cancer Registry, covering a population of about one million in southwest Germany, have often been used to estimate cancer incidence and survival for Germany in international studies such as EUROCARE [17]. For later years, data from selected federal states with sufficient data quality have been pooled in research studies [18] and by the German Centre for Cancer Registry Data (ZfKD) for national cancer statistics. 

For this study, incidence and mortality estimates were provided from the Saarland cancer registry for 1975–2015. National mortality (1980–2015) and incidence estimates (1999–2015) were provided from the ZfKD. National incidence estimates were calculated based on the incidence in regions with high completeness and estimated incidence based on a mortality/incidence ratio method in regions with insufficient completeness.

For analyses on patient characteristics and survival, a dataset from the Saarland cancer registry was used for years of diagnosis 1974–1997. For 1998–2015, a pooled national dataset from the ZfKD was used, covering 49% of the total German population (39.8 million inhabitants in 2014). Data selection criteria are described in the Supplementary Methods. 

For the US, data from the Surveillance, Epidemiology, and End Results (SEER) 9 database were used for all analyses [19,20]. SEER-9 covers 9.4% of the US population (29.0 million inhabitants in 2010). 

Appendix A provides an overview on the included registries for analyses on patient characteristics and survival. Women aged ≥15 years or older with a first invasive BC (ICD-10: C50) and passive mortality follow-up until December 2015 were included. Cases notified by death certificate only (DCO) or autopsy only were excluded in all analyses on patient characteristics and survival.

### 2.2. Definition of Variables

Age was investigated for all patients aged ≥50 years and grouped following screening recommendations in Germany (50–69 and 70+ years) [15]. Morphology was grouped according to the WHO classification of breast tumors (Appendix A). [21] Stage groups (I, II, III, IV) according to the 6th edition of the Union internationale contre le cancer (UICC) tumor, node, metastasis (TNM) classification were derived from the T, N, and M classification according to the TNM edition used at diagnosis. For US data, the breast adjusted American Joint Committee on Cancer (AJCC) 6th stage classification was used, whose stage groups are comparable to the derived German stage groups. Due to the large proportion of missing stage information in earlier years in Germany (>50%), stage information was only used for patients diagnosed in 1998–2015, with multivariable imputation of missing values by chained equation (details in the Supplementary Methods).Comparisons of stage distributions and five-year relative survival (RS) estimates before and after multiple imputation are shown in Appendix A, respectively.

### 2.3. Statistical Analysis

Annual incidence and mortality estimates were computed for the populations of Saarland and Germany and the US for 1975–2015. Overall and truncated age-specific estimates were age-standardized using the US2000 standard population. Estimates for Saarland were averaged over three calendar years due to the small sample size. 

Patient and tumor characteristics for 1974–2015 (all patients included in the analyses) and 2013–2015 (most recent period) were explored. Trends in stage distributions (1998–2015) are presented. Period analysis [22] was used to calculate five-year age-standardized (International Cancer Survival Standard, five age groups) [23] and age-specific RS. RS was calculated as the ratio of observed survival in cancer patients divided by expected survival derived by the Ederer II method [24] using life tables stratified by age, gender, calendar period, country, and, for the US, race as obtained from the German Federal Statistical Office [25] and the SEER program [26]. 

For all periods between 1979–1983 and 2013–2015, five-year age-standardized and age-specific RS for Germany and the US are reported. For Germany, for the periods 1979–1983 to 1999–2003 the Saarland Cancer Registry dataset was used to estimate RS, while the pooled national dataset was used from 2004–2006 to 2013–2015. 

Five-year age-standardized RS stratified by stage were estimated for the periods 2004–2006 to 2013–2015. For the same periods, five-year RS estimates with additional standardization by stage were computed. The stage distribution in the US in 2013–2015 was used as weights for standardization.

All survival calculations were carried out with SAS software (version 9.3), using macros developed for period analysis. The US incidence and mortality rates were calculated using the software SEER*Stat. 

### 2.4. Ethics Approval and Consent of Participate

The study was conducted in accordance with the Declaration of Helsinki, and the protocol was approved by the Ethics Committee of the Medical Faculty of Heidelberg (S-476/2013)

## 3. Results

### 3.1. Trends in Incidence and Mortality 

Age-standardized BC incidence was generally higher in the US than in Germany (Figure 1). Incidence increased steadily in Germany from 76.7 to 108.4 per 100,000 women per year between 1975 and 2008 followed by a steep increase until 2009 (129.0) and a decrease afterward (2015: 114.3). In contrast, in the US, incidence increased steeply at the beginning of the 1980s, then increased slightly in the years afterward until 1999, then decreased strongly until 2003 followed by a slight increase to 131.4 in 2015.

Incidence for persons aged 50–69 years was lower in Germany compared to the US in the early years, became comparable or higher in 2008–2012 but was 12.0% lower in 2015. Persons aged 70+ years had a lower BC incidence in Germany in all years with a 19.9% lower incidence in 2015. 

Age-standardized BC mortality decreased strongly in the US after 1989 from 33.2 to 19.4 (2015) per 100,000 persons and in Germany after 1996 from 33.4 to 25.3 per 100,000 persons. In 2015, BC mortality was 30.4% higher in Germany than in the US. For persons aged 50–69 years, BC mortality was almost comparable in earlier years but started to decrease earlier in the US than in Germany leading to 19.2% units higher mortality in Germany in 2015. For persons aged 70+ years, trends have diverged since 1989 when mortality started to decrease continuously in the US but not in Germany. After 1996, mortality also decreased in Germany, but only until 2007, and was followed by a slight increase. Therefore, BC mortality was 44.9% higher in Germany than in the US in 2015.

### 3.2. Patient Characteristics in 2013–2015

Overall, 512,534 and 671,095 BC patients remained for analyses after exclusion of 33,102 (6.1%) and 4847 (0.7%) DCO cases in Germany and the US, respectively (Appendix A; patient characteristics are described in Appendix A). In total, 93,721 and 68,776 BC patients from Germany and the US with a BC diagnosis in 2013–2015 were included (Table 1). The mean age at diagnosis was 63.7 and 61.9 years in Germany and the US, respectively. Invasive carcinoma of no special type was the most common morphology (74.0% and 72.6%) followed by invasive lobular carcinoma (12.8% and 10.3%). While most tumors were moderately differentiated in both countries (53.3% and 41.8%), fewer tumors were well differentiated in Germany, (13.9% vs. 23.7%). The upper-outer quadrant of the breast was the most common site of the tumor (34.7% and 33.1%). Tumors were more advanced in Germany and included 39.3% stage I and 7.6% stage IV tumors compared to 50.1% stage I and 5.1% stage IV tumors in the US.

### 3.3. Trends in Stage Distributions

Figure 2 shows stage distributions between 1998–2000 and 2013–2015 in Germany and the US by age at diagnosis. For German patients aged 50–69, a continuous shift to lower stages was observed with 36% and 49% stage I patients in 1998–2000 and 2013–2015, respectively. The change was particularly pronounced between 2004–2006 and 2007–2009. In the US, stage distributions were rather stable over time and, in 2013–2015, almost comparable to Germany. For patients aged 70+, stage distributions were stable over time in both countries with much more favorable stages in the US, with 54% compared to 28% stage I and 6% compared to 11% stage IV tumors in 2013–2015. 

### 3.4. Trends in Five-Year RS

In Germany, five-year age-standardized RS increased steadily between 1979–1983 (69.9%) and 2013–2015 (87.0%). A similar increase, but on a slightly higher level, was observed in the US (from 71.9% to 90.4%; Figure 3, Appendix A). For patients aged 50–69 years, RS was substantially lower in Germany than the in US in earlier years but comparable in 2013–2015 (≈91%). In contrast, for patients aged 70+ years, survival was comparable in earlier years but was 8.5% units higher in the US in 2013–2015.

Stage-specific trends in age-standardized five-year RS between 2004–2006 and 2013–2015 are shown in Figure 4 and Appendix A. For stages I, II, and III, RS increased by +1.3, 2.3, and +3.6% units in Germany and by +1.5, +2.2, and +5.5% units in the US, while no improvements were observed for stage IV patients. RS was comparable in both countries for stage I patients, but higher in Germany for stage II–IV patients with the largest difference for those with stage IV (+6.5% units in 2013–2015). 

After standardization for stage, age-standardized and age-specific RS still increased in Germany and the US between 2004–2006 and 2013–2015 (Figure 4, Appendix A). The previously described lower RS for patients aged 70+ years in Germany was no longer observed. 

## 4. Discussion

Our study provides a comprehensive comparison of long-term BC incidence, mortality, stage distribution, and RS trends in Germany and the US. In the US, incidence rates were much higher in all periods, whilst mortality rates have been lower since 1990. These differences were most pronounced for women aged 70+ years with 45% higher mortality in Germany in 2015. For this age group, very large differences in stage distributions were observed, with 28% stage I patients in Germany compared to 54% in the US. Trends in five-year RS revealed steady and strong improvements in RS in both countries but better prognosis in the US, particularly for patients aged 70+ years in more recent years. These country differences disappeared after accounting for the more favorable stage distribution in the US.

The utilization of mammography screening has a strong impact on BC incidence, stage distributions, mortality, and survival. Incidence increases with higher utilization, first due to early detection, but in both the short and long term also due to overdiagnosis [8,27]. Accordingly, the proportion of early stage cancers immediately increases after uptake and, in the long run, the incidence of advanced stage tumors should decrease. However, this effect was not consistently observed in observational studies [28,29]. Higher utilization of mammography screening should result in reductions of mortality. A meta-analysis of randomized trials reported a 35% mortality reduction in women invited to screening aged 50-69 years [8]. However, methodological concerns have been raised and results from observational studies remain contradictory [5,8,29]. Population-based survival should increase with increasing utilization of mammography screening because of earlier detection of tumors, overdiagnosis of less aggressive tumors, and due to added lead-time. Thus, it is necessary to consider differences in screening utilization when interpreting country differences in epidemiological data.

In the US, opportunistic mammography screening has become available since the 1980s and was promoted strongly by advocacy groups and federal and state agencies [8]. Health insurance reimbursement for mammography screening was already mandated in one state (Illinois) in 1981 and by 2000 in all remaining states but one (Utah). In Germany, organized screening was introduced between 2005 and 2009 (Saarland: 2006) and was less strongly publicly promoted. Among patients enrolled in the Kaiser Permanente Northwest health plan in the US, less than 5% of women aged 45 years or older had a screening mammography before 1982, but proportions increased rapidly to approximately 25% in 1986, 48% in 1991 and 75–79% between 1993 and 2006 [30]. Comparable data on utilization of mammography screening in the 1980s and 1990s in Germany are not available [31] but it was estimated that 25% of women aged 40–70 years had regular mammography screenings in the 1990s [32]. Results from a population-based study in Saarland show that in 2002–2004 45% of women aged 55 years or older reported having had a mammography within the last two years [33]. After the implementation of organized mammography screening in Germany, participation rates were more comparable in both countries, at least for the screening age group in Germany (50–69 years). According to the National Health Interview Survey in the US, in 2016, 68% of the women aged 50+ years and 52% of women age 75+ years had a mammography in the last two years [34]. In Germany, participation rates in the organized screening program were 51% in women aged 50–69 years in 2016 [35], but according to a representative population survey conducted in 2014/15, 74% of women in this age group reported having had a mammography (diagnostic or screening purpose) in the last two years [36]. From German claims data, it was estimated that far less than 20% of women age 70+ years had diagnostic mammography in 2014–2016 [37]. 

While we cannot provide any direct evidence for the impact of mammography screening, our results correspond to these country differences in screening utilization. We observed a much higher incidence in the US, particularly for elderly women. Furthermore, we saw a steeper increase in the US between 1980 and 1999. Also in line with higher screening utilization were the more favorable stage distributions in the US. Here, the largest differences were again observed for elderly patients with 54% stage I patients in the US compared to 28% in Germany. The introduction of organized screening mammography in Germany in 2005–2009 is in line with the rapid increase of BC incidence and of the proportion of stage I patients during this time [29]. 

BC mortality started to decline later in Germany than in the US, but prior to the introduction of the screening program. The decline was strongest for patients aged 50–69 years. A major difference across countries was that among women aged 70+ years, mortality has been continuously declining in the US since 1989, whereas the decline was less pronounced in Germany and was followed by a slight increase in recent years. Consequently, BC mortality in 2015 was 45% higher in Germany than in the US in 2015. Five-year RS was 9% units higher in patients aged 70+ years in the US in 2013–2015, but this difference can be explained by the less favorable stage distribution in Germany. Thus, the lower mortality observed in the US in elderly women can probably not be attributed to better BC care. One potential reason for the difference might be the less extensive use of screening mammography in elderly patients in Germany, but mammography screening is not recommended in this age group, as evidence on its benefit is lacking [16]. Another reason might be that elderly patients in the US are more likely to have a history of mammography screening during screening ages, which could impact mortality in later years. Investigations of trends by birth cohort, age, and calendar period might be a valuable approach to get further insights. Such an analysis has been conducted in Germany, but it was limited by using data from 1999–2008 only [38]. Thus, to further understand and finally overcome the higher mortality in elderly patients in Germany, further studies focusing on this population are needed and potentially should include data from other European countries to allow a more comprehensive international comparison.

While our results are in line with an earlier and more extensive uptake of mammography screening in the US, other country differences that might explain the findings require discussion. Regarding screening, clinical breast examination and breast self-examination are further available screening options. According to the latest evaluation of the International Agency for Research on Cancer (IARC), the evidence for an effect on breast cancer mortality is inadequate for both of the aforementioned screening methods [8]. Thus, even if the utilization of these methods might differ between Germany and the US, it is unlikely that they explain the country differences in mortality. Another potential explanatory factor for lower mortality in the US would be better cancer care in the US. However, we were not able to investigate the provision of cancer care, as only limited information on administered treatments was available in the cancer registry datasets. To our knowledge, the pattern of provision of major BC treatment approaches in Germany and the US have not yet been compared. However, it has been shown that available national BC treatment guidelines were quite comparable in both countries [14]. Furthermore, if BC care was better in the US than in Germany, stage-standardized relative survival should have been higher in the US. Thus, we argue that the observed differences may result from observed differences in the utilization of mammography screening in both countries. However, we cannot rule out that differences in the administration of cancer treatments or distribution of tumor biological factors as well as residual confounding may have played a role here.

Our study has several strengths and limitations. We used high-quality population-based cancer registry and official cause of death data to provide a joint interpretation of trends in incidence, mortality, RS, and stage distributions in Germany and the US. Limitations of our study were the lack of reliable stage information before 1998 and of detailed information on relevant tumor characteristics (e.g., hormone receptor, HER2/neu status) and administered cancer treatments. We did not conduct stratified analyses by ethnicity in the US, as ethnicity is related to socioeconomic status and these factors cannot be disentangled in the databases available for this analysis. Furthermore, differences in registration procedures and data quality issues in Germany and the US (e.g., utilization of different multiple primary cancer rules, higher DCO proportions in Germany) have to be considered [18]. It may be expected that RS estimates are slightly overestimated in Germany, especially for elderly patients, due to the exclusion of a higher proportion of DCO cases. Another limitation is that the datasets did not cover the whole country and, for SEER, it has been shown that the data include higher proportions of foreign-born persons, racial and ethnic minorities, and urban dwellers and, therefore, might not be fully representative for the entire US population [39]. Furthermore, differences in the official coding of cause of death in Germany and the US cannot be ruled out. 

## 5. Conclusions

Our study showed that differences in incidence, mortality, and RS between the US and Germany mostly decreased over time for patients aged 50–69 years. However, strong and increasing differences in BC mortality were observed in women aged 70+ years, resulting in 45% higher mortality in Germany in 2013–2015. Our results do not suggest differences in cancer care, as stage-standardized survival was comparable in both countries. The observed differences might rather be related to the later uptake and less extensive utilization of mammography screening in Germany. Further studies are warranted to fully understand and overcome the comparably high BC mortality in the rapidly growing elderly population in Germany.

## Figures and Tables

**Figure 1 cancers-12-02419-f001:**
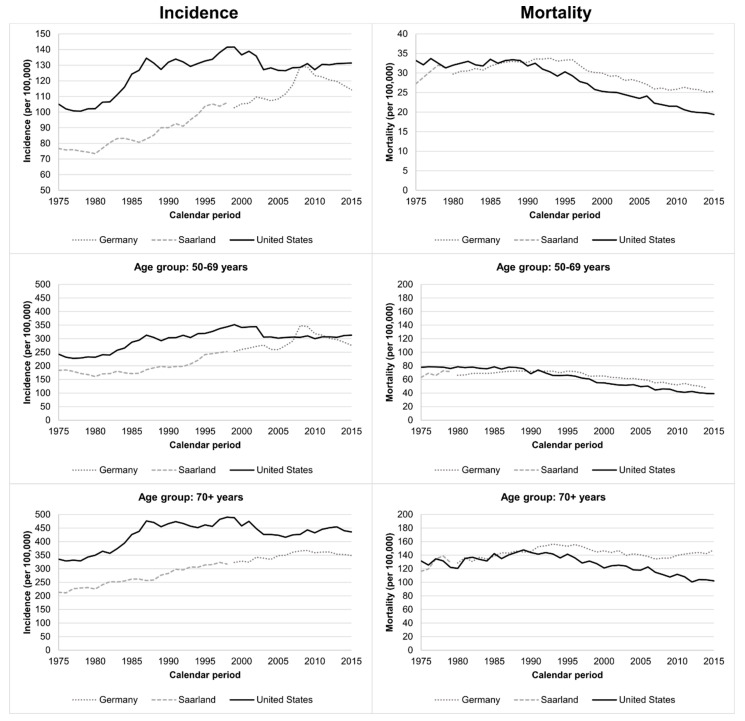
Age-standardized and age-specific breast cancer incidence and mortality trends in Germany/Saarland and the United States.

**Figure 2 cancers-12-02419-f002:**
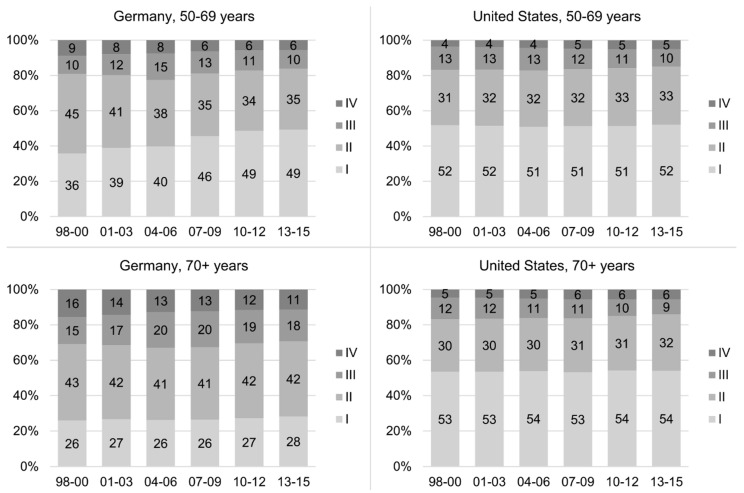
Trends in the distribution of stage (Union internationale contre le cancer (UICC)/ American Joint Committee on Cancer (AJCC), 6th edition) in Germany and the United States by age at diagnosis.

**Figure 3 cancers-12-02419-f003:**
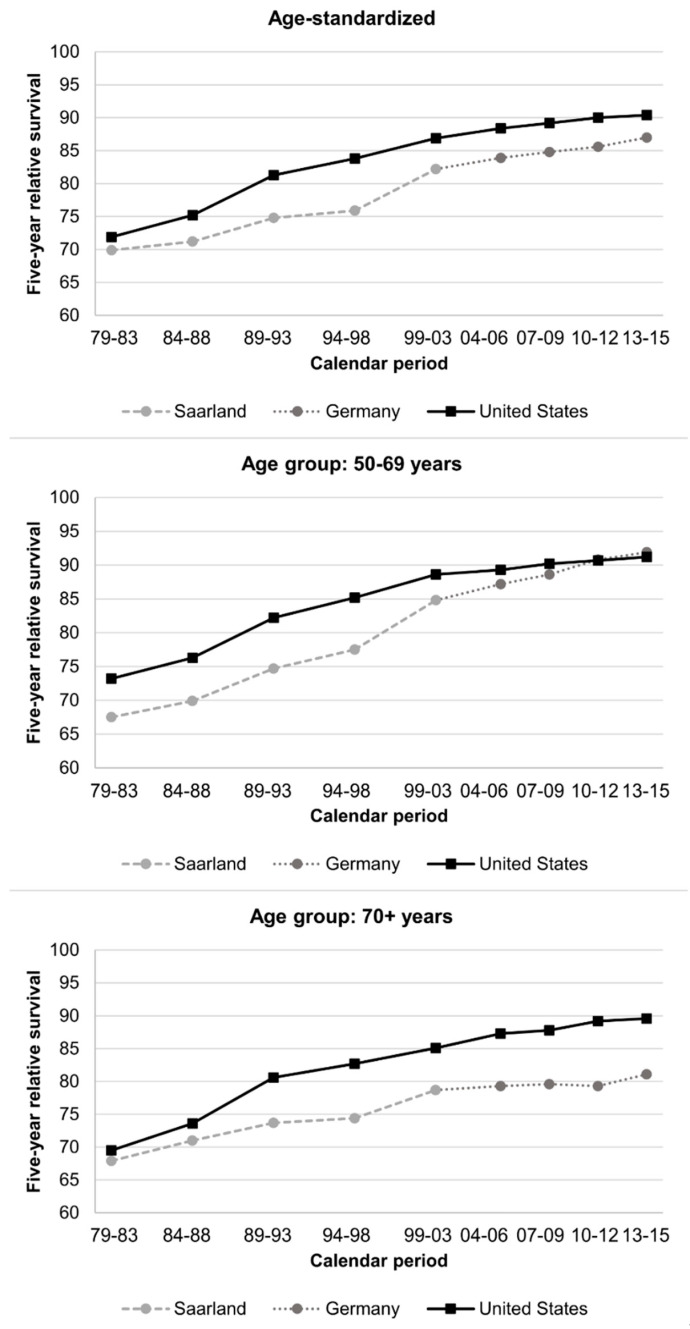
Trends in age-standardized and age-specific five-year relative survival in Germany, Saarland and the United States.

**Figure 4 cancers-12-02419-f004:**
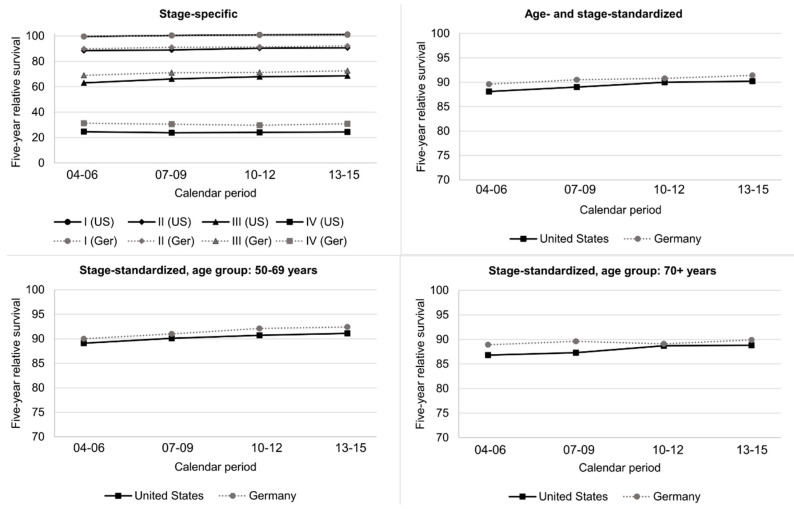
Trends in five-year stage-specific survival and five-year age-standardized and age-specific relative survival after standardization by stage (Union internationale contre le cancer (UICC)/American Joint Committee on Cancer (AJCC), 6th edition) in Germany and the United States.

**Table 1 cancers-12-02419-t001:** Patient and tumor characteristics of patients with a first invasive breast cancer diagnosed in Germany and the United States in 2013–2015.

Factor	Germany *N* (%) ^b^	United States *N* (%) ^b^
***N***	93,721	68,776
**Excluded DCO cases**	4209 (4.3)	360 (0.5)
**Age at diagnosis**		
15–29	455 (0.5)	380 (0.6)
30–39	3111 (3.3)	2819 (4.1)
40–44	4482 (4.8)	3910 (5.7)
45–49	8609 (9.2)	6080 (8.8)
50–54	11,762 (12.6)	7870 (11.4)
55–59	9851 (10.5)	8438 (12.3)
60–64	11,511 (12.3)	9449 (13.7)
65–69	10,577 (11.3)	9635 (14.0)
70–74	10,371 (11.1)	7502 (10.9)
75–79	10,994 (11.7)	5415 (7.9)
80+	11,998 (12.8)	7278 (10.6)
Mean (standard deviation)	63.7 (13.9)	61.9 (13.6)
**Morphology**		
Invasive carcinoma of no special type	69,376 (74.0)	49,938 (72.6)
Pleomorphic carcinoma	2004 (2.1)	3621 (5.3)
Invasive lobular carcinoma	12,033 (12.8)	7106 (10.3)
Tubular carcinoma	875 (0.9)	326 (0.5)
Mucinous carcinoma	1590 (1.7)	1319 (1.9)
Medullary carcinoma	326 (0.3)	103 (0.1)
Invasive micropapillary carcinoma	180 (0.2)	425 (0.6)
Metaplastic carcinoma of no special type	430 (0.5)	333 (0.5)
Invasive papillary carcinoma	437 (0.5)	280 (0.4)
Other/not specified	6470 (6.9)	5325 (7.7)
**Grade**		
Well-differentiated (I)	13,024 (13.9)	16,273 (23.7)
Moderately differentiated (II)	49,917 (53.3)	28,737 (41.8)
Poorly differentiated (III)	26,386 (28.2)	19,810 (28.8)
Unknown	4394 (4.7)	3956 (5.8)
**Tumor site (ICD-10 code C50.X)**		
Nipple and areola (0)	995 (1.1)	337 (0.5)
Central portion (1)	4234 (4.5)	3079 (4.5)
Upper-inner quadrant (2)	10,186 (10.9)	8590 (12.5)
Lower-inner quadrant (3)	5281 (5.6)	3873 (5.6)
Upper-outer quadrant (4)	32,520 (34.7)	22,774 (33.1)
Lower-outer quadrant (5)	7355 (7.8)	5245 (7.6)
Axillary tail (6)	173 (0.2)	352 (0.5)
Overlapping sites (7)	13,008 (13.9)	15,794 (23.0)
Unspecified site (9)	19,969 (21.3)	8732 (12.7)
**Stage according to UICC/AJCC** ^a^		
I	36,708 (39.3)	34,415 (50.1)
II	36,758 (39.4)	23,598 (34.4)
III	12,777 (13.7)	7119 (10.4)
IV	7133 (7.6)	3521 (5.1)

DCO = death certificate or autopsy only case; ICD = International Classification of Disease; ^a^ Patients with stage 0 were excluded (*N* = 345 (0.4%) (Germany) and *N* = 123 (0.2%) (United States); ^b^ Percentage of the number of patients in each category in relation to the total number of patients in this country (numbers might not add up to 100% due to rounding). For age, the mean and standard deviation are shown in the line “mean (standard deviation)”.

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
