# Peer review of "Divergent Patterns and Trends in Breast Cancer Incidence, Mortality and Survival Among Older Women in Germany and the United States"

_cancers, 2020, doi:10.3390/cancers12092419_

Round 1
Reviewer 1 Report
The manuscript “Divergent patterns and trends in breast cancer incidence, mortality and survival among older women in Germany and the United States” by Jansen et al. describes the differences in breast cancer incidence, its related mortality and relative survival between the women populations of the United States and Germany. The manuscript is properly constructed and well written. Although, there are several limitations of the study f. ex. lack of information about subtypes of breast cancers diagnosed or patient ethnicity, which authors correctly list at the end of discussion section, the presented analysis still provides valuable information and should meet the interest of the readers of Cancers. Below I am addressing some minor comments for authors to include in order to improve the quality of their work.
- The authors sometimes present time period including total year f. ex. 2013-2015, and sometimes abbreviation: 2013-15. Please choose one method and be consistent.
- Please add the title and list the authors on the first page of supplementary material.
Author Response
The manuscript “Divergent patterns and trends in breast cancer incidence, mortality and survival among older women in Germany and the United States” by Jansen et al. describes the differences in breast cancer incidence, its related mortality and relative survival between the women populations of the United States and Germany. The manuscript is properly constructed and well written. Although, there are several limitations of the study f. ex. lack of information about subtypes of breast cancers diagnosed or patient ethnicity, which authors correctly list at the end of discussion section, the presented analysis still provides valuable information and should meet the interest of the readers of Cancers. Below I am addressing some minor comments for authors to include in order to improve the quality of their work.
Re: We would like to thank the reviewer for the appreciation of our work.
- The authors sometimes present time period including total year f. ex. 2013-2015, and sometimes abbreviation: 2013-15. Please choose one method and be consistent.
Re: We have now chosen the abbreviated version and use it consistently in the manuscript and supplementary material.
- Please add the title and list the authors on the first page of supplementary material.
Re: Done as suggested.
Reviewer 2 Report
The authors explored patterns in older cohort of breast cancer patients in Germany and the US. They used Five-year interval survival to analyze the mortality/survivability of the patients. the methods seem fine to me and the results are well presented. However, I have a minor concern:
- The study should mention the recent approaches for survivability study such as https://doi.org/10.3389/fgene.2019.00256 and https://doi.org/10.1007/s13721-020-00249-4
- The study assumes that all BC are the same disease, neglecting the heterogeneity nature of the disease/subtype. a small explination in the introduction will increase the interest of the experts for this work.
Author Response
The authors explored patterns in older cohort of breast cancer patients in Germany and the US. They used Five-year interval survival to analyze the mortality/survivability of the patients. the methods seem fine to me and the results are well presented. However, I have a minor concern:
- The study should mention the recent approaches for survivability study such as https://doi.org/10.3389/fgene.2019.00256 and https://doi.org/10.1007/s13721-020-00249-4
Re: We thank the reviewer to pointing us to these two interesting studies and added them in the introduction (page 3, line 76-78, reference: 11 & 12).
- The study assumes that all BC are the same disease, neglecting the heterogeneity nature of the disease/subtype. a small explination in the introduction will increase the interest of the experts for this work.
Re: We agree with the reviewer and now explain that breast cancer is a heterogeneous disease with strong differences in prognosis (page 3, lines 74-76).
Reviewer 3 Report
Since the focus is on Germany and US, I would suggest including statistics from these two countries within the very first paragraph. Please include these 3 references – PMIDs: 26933878, 31456176 and 31912902
It is not made very clear why Germany and US are the focus countries. Is it because authors are interested in numbers from their country, Germany and US happens to be the country with most published statistics?
The third paragraph in Introduction opens with a premise for comparisons across ‘many’ countries but quickly defines the scope of the article in the very next statement. Please try editing that first sentence.
Is it necessary to show ‘Saarland’ in the Figures. Cannot this be simply Germany and the details can be provided in the Methods and in the Figure Legends?
Is the data provided in Table absolute numbers? Or are they numbers per certain fractions?
Do authors have some data to share on the receptor positive or the TNBCs ?
The study seems to focus only on mammography? Why ? What other major differences in screening of breast cancer patients is evident, when comparing Germany with US?
In continuation of my above comment, how about the major differences in the clinical management of breast cancer patients in the two countries. Please discuss.
Author Response
Since the focus is on Germany and US, I would suggest including statistics from these two countries within the very first paragraph.
Re: We would like to thank the reviewer for his positive evaluation. As suggested, we added statistics from Germany and the US in the introduction (case/death numbers and 5-year relative survival rates) (page 3, lines 63-65 & 71-74).
Please include these 3 references – PMIDs: 26933878, 31456176 and 31912902
Re: As suggested by the reviewer, the three references are now included in the manuscript (Siegel et al. (2020): page 3, line 64 & 74; Ahmad (2019): page 3, line 68; Berkemeyer et al (2016): page 10, lines 302-304).
It is not made very clear why Germany and US are the focus countries. Is it because authors are interested in numbers from their country, Germany and US happens to be the country with most published statistics?
Re: We have chosen the US as second country for several reasons. We were interested in very long-term trends and, thus, needed another country with high-quality cancer registration since the 1970s. In addition, the cancer registry had to provide information on cancer stage. Furthermore, we wanted to choose a country with similar treatment guidelines but with a key difference in mammography screening utilization. It has been reported that treatment guidelines were similar in Germany and the US (Wolter et al, 2012), whereas mammography screening started much earlier in the US and had been utilized more extensively. We now directly state the rational for choosing the US for comparison (page 3, lines 83-94).
The third paragraph in Introduction opens with a premise for comparisons across ‘many’ countries but quickly defines the scope of the article in the very next statement. Please try editing that first sentence.
Re: We have followed the reviewers suggestion and edited this sentence (page 3, line 83-85).
Is it necessary to show ‘Saarland’ in the Figures. Cannot this be simply Germany and the details can be provided in the Methods and in the Figure Legends?
Re: While we use the numbers of the federal state Saarland representatively for Germany, we would like to indicate in the Figures the data source of each estimate. Otherwise, readers might be confused by finding estimates from “Germany” for calendar years where cancer registration had not been established nationwide in Germany. Furthermore, some short-term trends in incidence/mortality might be wrongly interpreted without this differentiation, as the incidence/mortality difference between two periods were sometimes based on a switch from Saarland to German estimates (although Saarland and Germany are mostly comparable).
Is the data provided in Table absolute numbers? Or are they numbers per certain fractions?
Re: In Table 1 and Supporting Table 5, the number of cases and, in brackets, the percentage of patients in the category are shown. We have now added an explanation of the percentage in the footnote. (Table 1, footnote b, lines 196-198). In Supporting Table 3, we added “%” after the numbers. In Supporting Tables 4, 6, and 7, we now clarify that standard errors are shown in brackets.
Do authors have some data to share on the receptor positive or the TNBCs ?
Re: Unfortunately, we do not have information on molecular characteristics/subtypes in the German database. Clinical cancer registration has only recently been started in Germany. For the SEER data, breast cancer subtype is available, but only since 2010. This limitation of our study is described in the discussion (page 11, lines 328-330)
The study seems to focus only on mammography? Why ? What other major differences in screening of breast cancer patients is evident, when comparing Germany with US?
Re: We focus our manuscript on the utilization of mammography, as the difference in the utilization between Germany and the US is, in our opinion, very strong. Furthermore, as the organized mammography screening program has only been recently started in Germany (2005-2009), evaluation of (long-term) trends in incidence, mortality and survival in screening age groups is of particular concern. Other breast cancer screening methods are clinical breast examination and breast self-examination. We now discuss these two alternative screening methods in the manuscript (page 11, lines 310-314).
In continuation of my above comment, how about the major differences in the clinical management of breast cancer patients in the two countries. Please discuss.
Re: We agree with the reviewer that it is important to compare clinical management of breast cancer in Germany and the US. However, both cancer registry datasets do not provide sufficient, good-quality information on treatment. While we are not aware of a study comparing breast cancer treatment in the US and Germany on the population-level, it has been reported that treatment guidelines are comparable. Furthermore, if the lower mortality rates in the US were caused by better cancer care in the US, the stage-standardized relative survival should have been better in the US. We now discuss these aspects in the manuscript (page 11, lines 315-325).
Round 2
Reviewer 3 Report
All of my concerns have been adequately addressed.